# Cardiovascular Disease in Diabetes and Chronic Kidney Disease

**DOI:** 10.3390/jcm12226984

**Published:** 2023-11-08

**Authors:** Sowmya Swamy, Sahibzadi Mahrukh Noor, Roy O. Mathew

**Affiliations:** 1Department of Medicine, School of Medicine, George Washington University, Washington, DC 20052, USA; 2Department of Medicine, School of Medicine, Loma Linda University, Loma Linda, CA 92350, USA; 3Department of Medicine, Loma Linda VA Healthcare System, 11201 Benton Street, Loma Linda, CA 92357, USA

**Keywords:** diabetes mellitus, chronic kidney disease, cardiovascular disease, epidemiology, sglt2 inhibitors

## Abstract

Chronic kidney disease (CKD) is a common occurrence in patients with diabetes mellitus (DM), occurring in approximately 40% of cases. DM is also an important risk factor for cardiovascular disease (CVD), but CKD is an important mediator of this risk. Multiple CVD outcomes trials have revealed a greater risk for CVD events in patients with diabetes with CKD versus those without. Thus, reducing the risk of CKD in diabetes should result in improved CVD outcomes. To date, of blood pressure (BP) control, glycemic control, and inhibition of the renin-angiotensin system (RASI), glycemic control appears to have the best evidence for preventing CKD development. In established CKD, especially with albuminuria, RASI slows the progression of CKD. More recently, sodium glucose cotransporter 2 inhibitors (SGLT2i) and glucagon-like peptide receptor agonists (GLP1RA) have revolutionized the care of patients with diabetes with and without CKD. SGLT2i and GLP1RA have proven to reduce mortality, heart failure (HF) hospitalizations, and worsening CKD in patients with diabetes with and without existing CKD. The future of limiting CVD in diabetes and CKD is promising, and more evidence is forthcoming regarding combinations of evidence-based therapies to further minimize CVD events.

## 1. Introduction

Diabetes mellitus (DM) and chronic kidney disease (CKD) are two of the strongest risk factors for cardiovascular disease (CVD) [1]. This includes every aspect of CVD, including atherosclerotic CVD (ASCVD), heart failure (HF), valvular disease, peripheral arterial disease (PAD), stroke, and arrhythmias [2]. In addition, overall and CVD-related mortality remain high for these conditions. The co-existence of these conditions has additive and, at times, multiplicative effects on these outcomes [3]. The following review summarizes the current epidemiology of the individual and the combined effects of DM and CKD on CVD outcomes. A review of the novel therapies that mitigate this risk is provided.

## 2. Epidemiology

In 2021, diabetes was the eighth leading cause of death in the United States [4,5]. DM is a well-known risk factor for ASCVD and is a leading cause of morbidity and mortality in patients with DM [6,7,8]. The Framingham study demonstrated a two- to three-fold increased risk of clinical atherosclerotic disease in patients with DM [8]. In a meta-analysis, patients with DM had a higher risk of ASCVD (Hazard Ratio [HR] 1.30, *p* < 0.0001), congestive HF (HR 1.44, *p* < 0.0001), kidney replacement therapy (HR 2.52, *p* < 0.0001), and death (HR 1.21, *p* < 0.0001) compared to those without DM. Patients with DM are more than twice as likely to die from cardiovascular disease (CVD) compared to those without DM, and CVD accounts for approximately 60% of the life years lost from diabetes [9]. Patients with DM not only have a higher incidence of cardiovascular (CV) events but also have worse CVD outcomes [10].

Given these findings, DM has been recognized as a “CVD equivalent” for over two decades [11]. An observational study among the Finnish population showed that the risk of death from coronary heart disease (CHD) in patients with DM and no history of myocardial infarction (MI) was not statistically different from persons without DM but with prior MI even when adjusted for factors such as age, sex, total cholesterol, hypertension, and smoking (HR 1.2–1.5, 95% Confidence Interval (CI) 0.7–2.6). In a separate study, the relative risk (RR) for CHD and CV death was 2.82 (95% CI 1.85 to 4.28) in men with DM only, 3.91 (95% CI 3.07 to 4.99) in men with MI, and 8.93 (95% CI 6.13 to 12.99) in men with both DM and CHD/MI. Case fatalities among men with diabetes were similar to those with prior CHD. CHD and CVD-related mortality increased with the increasing duration of diabetes, with the risk eventually approaching that of patients with MI without diabetes [12].

On the other hand, newer studies have questioned whether the presence of diabetes is necessarily a risk equivalent for future ASCVD [13,14,15,16]. The older studies mainly included MI as an endpoint and did not include PAD or stroke as part of CVD or ASCVD. Also, there was a paucity of data regarding ethnically diverse populations, women, and younger patients. A recent study pooled data from four diverse prospective cohort studies and compared the risk of a broadly defined CVD endpoint among patients with DM and without prior CVD (DM+/CVD−) versus patients without DM and with CVD (DM−/CVD+) [1]. Compared to DM−/CVD+ patients, those with DM+/CVD− experienced 14% lower CVD risk (HR 0.86, 95% CI: 0.8–0.9) and 24% lower ASCVD risk (HR: 0.76, 95% CI: 0.69–0.84) than DM−/CVD+ patients. This finding, which is contrary to the previously cited data, was primarily driven by the duration and level of control of DM. If the duration of DM was ≥10 years, the fully adjusted HR (95% CI) was 1.20 (1.06–1.35) as compared to DM−/CVD+, and if the baseline hemoglobin A1c (HbA1c) was ≥9%, the HR (95% CI) was 1.34 (1.18–1.52). The presence of CVD risk equivalent diabetes (longer duration and poorer control) was associated with higher event rates compared to non-CVD risk equivalent diabetes (CVD: 53.8 vs. 29.2 per 1000 person-years; ASCVD: 27.7 vs. 14.1 per 1000 person-years). This study suggests that DM alone does not equate to CVD risk equivalence, and several factors affect the magnitude of CVD risk conferred by DM. Also, since the data were gathered from studies carried out at least two decades ago, more aggressive treatment strategies and recent pharmacological advancements in therapies may have further mitigated CVD risk due to DM. The current American College of Cardiology (ACC)/American Heart Association (AHA) guidelines now, accordingly, recommend high-intensity statins among patients with DM only when 10-year risk is ≥20%, as compared to moderate-intensity statins for all patients with DM [17]. Risk-enhancers include duration ≥ 10 years for type 2 DM and 20 years for type 1 DM, urine albumin/creatinine ≥ 30 mcg/mg, estimated glomerular filtration rate (eGFR) < 60 mL/min/1.73 m^2^, retinopathy, neuropathy, or arterial-brachial-index (ABI) < 0.9.

CKD develops in 30–40% of those with type 1 DM and 40% of those with type 2 DM. The presence of CKD in diabetes acts as a risk multiplier for CVD [18,19,20]. In the Action to Control Cardiovascular Risk in Diabetes (ACCORD) trial, the presence of CKD DM was associated with a 41% increased risk (*p* = 0.02) for CVD composite primary outcome and a 39% increased risk (*p* = 0.04) for all-cause mortality as compared to DM alone [21]. It should be noted that participants in the ACCORD trial had mild CKD, as individuals were excluded if they had a serum creatinine > 1.5mg/dL. This stresses the importance of CKD in DM, even at relatively preserved eGFR.

A post-hoc analysis of the ACCORD trial showed that in patients with diabetes and mild-to-moderate CKD, the risk of having the primary composite outcome was 87% higher (HR 1.86, 95% CI 1.6–2.1, *p* < 0.001), the risk of all-cause mortality was 97% higher (HR 1.97, 95% CI 1.70–2.28, *p* < 0.0001), the risk for cardiovascular mortality was 119% higher (HR 2.18, 95% CI 1.75–2.72, *p* < 0.0001), and the risk for fatal or nonfatal congestive HF was 219% higher than patients without CKD [3]. In the Action in Diabetes and Vascular Disease: Preterax Controlled Evaluation (ADVANCE) trial, an annual decrease in eGFR slope was significantly associated with increased risk of the primary outcome (composite of major renal events, major macrovascular events, and all-cause mortality), HR 1.30 (95% CI 1.17, 1.43) compared with a stable change in eGFR. An annual increase in eGFR had demonstrated trends toward lower hazards for the primary, but the estimate included the possibility of no effect: HR 0.96 (95% CI 0.86, 1.07) [22]. Thus, the eGFR slope in type 2 DM has been utilized as a surrogate endpoint for kidney outcomes as well as a prognostic factor for cardiovascular disease and all-cause mortality.

The weaker association between CKD and CVD in ACCORD and ADVANCE highlights the importance of the severity of CKD in considering risk. In both trials, the majority of the patients had mild CKD at baseline. Several observational studies have demonstrated a greater association between more advanced stages of CKD and CVD. In a Chinese prospective study in patients with DM, the rate of new cardiovascular endpoints (cardiovascular death, new admissions due to angina, MI, stroke, revascularization, or HF) increased from 2.6% (2.0–3.3) to 25.3% (15.0–35.7) (*p* < 0.001) from stage 1 (eGFR > 90 mL/min/1.73 m^2^) to stage 4 (eGFR < 30 mL/min/1.73 m^2^) CKD, representing a tripling of risk in stage 4 relative to stage 1. The respective all-cause mortality rate increased from 1.2% (95% CI 0.8–1.7) to 18.3% (95% CI 9.1–27.5) (*p* < 0.001). Compared with patients with stage 1 CKD with normal urine albumin, stage 4 CKD and albuminuria at baseline conferred a 16-fold increased risk for all-cause mortality and cardiovascular endpoints [23]. The European FIELD study showed that lower eGFR versus eGFR ≥ 90 mL/min/1.73 m^2^ increased the risk for total CVD events with a HR of 1.14 (95% CI 1.01–1.29) for eGFR 60–89 mL/min/1.73 m^2^ and HR 1.59 (95% CI 1.28–1.98) for eGFR 30–59 mL/min/1.73 m^2^ (*p* < 0.001) [24].

Albuminuria is an independent predictor of ASCVD and HF in DM patients with CKD. The Casale Monferrato Study concluded that in type 2 DM, macroalbuminuria was the main predictor of mortality, independently of both eGFR and cardiovascular risk factors. HR for CV mortality in patients with AER 20–200 and >200 ug/min was 1.06 and 2.0, respectively (*p* < 0.0001). HR for CV mortality compared to CKD1 across CKD stages 2–4 was 0.65, 0.79, 0.67, and 2.03 (*p* = 0.27). In an analysis stratified by albuminuria categories, a significant increasing trend in risk with decreasing eGFR was evident only in people with macroalbuminuria [25]. However, in the FIELD study, both microalbuminuria and macroalbuminuria increased CVD risk, with HR 1.25 (95% CI 1.01–1.54) and 1.19 (95% CI 0.76–1.85), respectively (*p* = 0.001 for trend) with eGFR ≥ 90 mL/min/1.73 m^2^. In multivariable analysis, 77% of the effect of eGFR and 81% of the effect of albuminuria were accounted for by other variables, principally low HDL-cholesterol and elevated blood pressure [24]. A meta-analysis of 26 cohort studies looked at the association between albuminuria and coronary events. The presence of albuminuria was associated with a 50% increase in coronary risk (risk ratio 1.47, 95% CI 1.23–1.74). In those with macroalbuminuria, the risk was more than doubled (risk ratio 2.17, 1.87–2.52), supporting a continuous dose-dependent relationship [26]. These findings emphasize the importance of taking albuminuria along with eGFR into consideration when evaluating cardiovascular risk in patients with DM [27].

A Japanese study showed that the risk for CVD was associated with the progression of the albuminuria stage rather than the eGFR stage in DM patients, who are at relatively low risk for CVD. Compared with patients with no CKD as a reference, those with ACR ≥ 35.0 mg/mmol and co-existing eGFR < 89 mL/min/1.73 m^2^ showed increased risk for CVD onset, whereas those with eGFR ≥ 90 mL/min/1.73 m^2^ did not. Those with ACR < 3.5 mg/mmol and eGFR < 60 mL/min/1.73 m^2^ did not show any increased risk. Among patients with no CKD at baseline (eGFR ≥ 90 mL/min/1.73 m^2^ and albumin-to-creatinine [ACR] ratio < 3.5 mg/mmol [<30 mg/gm]), worsening of ACR to ≥3.5 mg/mmol was associated with an increased risk for CVD, whereas a decrease in eGFR to 90 mL/min/1.73 m^2^ was not associated with an increased CVD risk [28]. This suggests that at higher eGFR, albuminuria continues to carry important CVD prognostic information. In a post-hoc analysis of the ADVANCE study, patients with both UACR > 300 mg/g and eGFR < 60 mL/min per 1.73 m^2^ at baseline had a 3.2-fold higher risk for cardiovascular events and a 22.2-fold higher risk for kidney events, compared with patients with neither of these risk factors [29]. Another meta-analysis of 28 cohorts showed that albuminuria was also associated with an increased risk of end-stage kidney disease (ESKD), supporting its use as a surrogate endpoint for ESKD. The adjusted hazard ratio (HR) for a 30% decrease in albuminuria during a period of 2 years was 0·83 (95% CI 0·74–0·94) and conferred a more than 1% absolute reduction in the 10-year risk of ESKD. A 43% increase in albuminuria resulted in an increased risk of ESKD, HR 1.16 (95% CI 1.03–1.31) [30].

Due to the increased risk of ASCVD in patients with CKD concurrent with DM, CKD should be accounted for while evaluating risk and implementing prevention strategies for cardiovascular disease and mortality. Furthermore, consider eGFR and albuminuria as the essential markers for CKD when assessing the risk, prognosis, and treatment strategies of patients with DM. The Table 1 summarizes some of the pivotal studies highlighting the relationship between DM, kidney disease, and CVD risk.

## 3. Treatment Options

### 3.1. Hypertension/Albuminuria

Prevention of cardiovascular disease among patients with DM remains a central tenet of medical therapy. The distinction between type 1 and 2 DM is likely subtle, but more importantly, and unfortunately, the evidence base for CVD risk reduction in type 1 DM is less robust [31]. Blood pressure control is one of the primary targets for CVD risk reduction in people with diabetes. Blood pressure targets appear to be dependent on several factors. Among patients with DM and albuminuria or higher cumulative CVD risk per risk estimator such as the pooled cohort equation, lower blood pressure (<130/80) is recommended. In those without these high-risk features, a target of <140/90 may be trialed with further reductions based on individual tolerance. Other factors to be considered include age (younger age with likely greater tolerance for lower blood pressures) and preceding cardiovascular disease. To date, the ACCORD-blood pressure (BP) trial provides the best evidence for BP targets in patients with type 2 DM [32]. As compared to a “standard” BP target of <140/90, targeting BP < 120/80 did not result in reduced major cardiovascular events but was associated with lower stroke events, despite higher adverse events overall. In subsequent analysis, tighter control of blood pressure was associated with significant reductions in major cardiovascular events among those also randomized to the standard glycemic arm of the glycemic control portion of the trial [33]. This remains a speculative finding but is an important consideration as less stringent glycemic targets (≥7% Hemoglobin A_1c_) are the norm for type 2 DM based on the findings from ACCORD.

The choice of ideal agents for optimal risk reduction follows these general principles. Renin-Angiotensin System inhibitors (RASI) appear to provide the greatest benefit for CVD and kidney failure risk reduction among individuals with DM with baseline albuminuria, and particularly macro-albuminuria (>300 mg/gm [33.9 mg/mmol]). In the landmark trial by the Collaborative Study Group, Captopril (an angiotensin converting enzyme inhibitor—ACEI) was shown to have significant effects in slowing the rate of loss of kidney function and reducing the risk of death, or kidney replacement therapy, by half independent of its anti-hypertensive effects. Additionally, ACEI was found to demonstrate significantly less proteinuria [34]. Angiotensin receptor blockers (ARB) similarly showed effects in the Irbesartan Diabetic Nephropathy Trial (IDNT) and in the Reduction of Endpoints in NIDDM with the Angiotensin II Antagonist Losartan Study (RENAAL), both of which demonstrated lower risk for progression to ESKD and serum creatinine doubling time with the use of irbesartan or losartan, respectively [35,36].

In addition, albuminuria has been identified as a risk factor for cardiac and kidney outcomes in patients with type 2 diabetes [37]. RASI has demonstrated a reduction in baseline albuminuria when compared to placebo, with the groups with the greatest reduction in albuminuria showing the most significant reduction in risk for cardiac events [38]. While there may be differences between the main RASIs in use (ACEI and ARB), clinically, these agents are used interchangeably. On the other hand, clinical trial evidence has demonstrated the potential harm of combining ACEI and ARB for albuminuria reduction due to the risk of acute kidney injury and hyperkalemia [39,40].

### 3.2. Lipids

Hyperlipidemia is a common finding among patients with diabetes. Current guidelines recommend treatment with statin to achieve target LDL cholesterol of <100 mg/dL, with the goal of <70mg/dL in patients with concomitant DM and CVD. Hyperlipidemia is also associated with CKD, with dyslipidemia worsening as CKD progresses [41]. Statins are the first-line lipid lowering medication for patients with CKD and hyperlipidemia [42,43]. The use of statins in patients with CKD has been shown to decrease proteinuria, attenuate the progression of decline in kidney function, and reduce overall CVD risk [44,45]. Much of the cardiovascular risk associated with CKD is thought to be derived from persistent inflammation. Statins are thought to exert anti-inflammatory effects that explain their benefit beyond LDL lowering [46]. While the data for statin therapy in patients with CKD not on dialysis are robust, the data remains unclear for patients with kidney failure on dialysis. Two [47,48] landmark trials have demonstrated the lack of cardiovascular benefit of statins in patients on dialysis.

### 3.3. Glycemia and Glomerular Hyperfiltration

Sodium/glucose cotransporter 2 (SGLT2) inhibitors prevent glucose reabsorption in the proximal tubules of the kidney and lower the threshold for glucose excretions, leading to a negative sodium and water balance and an ultimate reduction in blood pressure [49,50]. Several landmark trials have demonstrated the cardioprotective effects of SGLT-2 inhibitors in patients with DM, particularly in reducing the rates of cardiovascular hospitalizations and deaths as related to HF [51,52,53,54,55]. CKD is an independent risk factor for the development of HF, with a cyclic effect in patients with HF leading to progressive kidney decline. SGLT2 inhibitors have noted renoprotection in patients with both normal and impaired GFR [56]. In identifying cardiovascular outcomes across various eGFRs, a systematic review by Arshad et al. analyzed six studies with SGLT-2 inhibitors, totaling a cohort of 56,869 patients, with 19,311 patients noted to have an eGFR < 60 mL/min/1.73 m^2^. Interestingly, there was no statistically significant reduction in major adverse cardiovascular events (MACE) in patients on SGLT-2 inhibitors with eGFR > 60 mL/min/1.73 m^2^. However, in patients with eGFR < 60 mL/min/1.73 m^2^, there was a 15% risk reduction in MACE [57]. When considering albuminuria, SGLT-2 inhibitors were associated with a reduction in HF-associated hospitalizations and deaths, regardless of albuminuria level at baseline, with the added benefit of halting the progression of albuminuria and even reversing it [58]. Additionally, SLGT-2 inhibitors also demonstrated the greatest amount of kidney and cardiovascular event reduction in those with severe albuminuria (UACR ≥ 3000 mg/g) [59].

Glucagon-like peptide-1 receptor agonists (GLP-1RA) are another class of medications that have gained favor in their use in treating patients with concomitant diabetes and kidney dysfunction by preventing the onset of albuminuria and reducing the decline in eGFR. GLP-1RA has effects on traditional risk factors for CKD, including hypertension, by inducing natriuresis and diuresis [60]. The efficacy and safety of this medication were first demonstrated in the landmark AWARD-7 trial, which tested dulaglutide in patients with moderate to severe CKD. The data were notable for better glycemic control in patients treated with dulaglutide, in addition to a slower rate of reduction in eGFR [61]. Data from the recently completed A Research Study to See How Semaglutide Works Compared to Placebo in People With Type 2 Diabetes and Chronic Kidney Disease (FLOW; NCT 03819153) is highly anticipated to provide additional kidney and cardiovascular benefits of GLP-1RA in patients with DM and CKD [62]. Thus, GLP-1RA is the agent of choice in patients with concomitant DM and CKD, as in addition to the ability to retard eGFR decline, it also reduces the risk of ASCVD events [63]. Additionally, some observational cohort studies demonstrate the glycemic benefits of GLP-1RA in patients with ESKD and transplantation, though larger clinical trials are necessary to further evaluate these findings. It is important to note that the cardiovascular benefits of GLP-1RA do not appear to be across the class, and that benefit is limited to certain agents. [64,65]

### 3.4. Cardiac Remodeling

Aldosterone is secreted in response to low sodium intake, a reduction in blood volume, or an increase in serum potassium levels. Aldosterone acts on the mineralocorticoid receptor (MR) in the distal nephron to cause reabsorption of sodium and potassium secretions. MRs are found in the vascular bed, inflammatory cells, and podocytes of the kidneys [66]. Overactivation of these receptors causes adverse effects, including fibrosis and chronic inflammation that can lead to chronic kidney disease and cardiovascular disease [67,68]. Finerenone is a novel, selective MR antagonist (MRA) associated with a reduced risk of cardiovascular events in patients with concomitant CKD and type 2 DM. In the Finerenone in Reducing Kidney Failure and Disease Progression in Diabetic Kidney Disease (FIDELIO-DKD) trial, 5734 patients with CKD and type 2 DM were randomized to receive placebo or finerenone. Results noted that treatment with finerenone resulted in a lower risk of CKD progression and cardiovascular events than placebo [69]. Of note, a dose-dependent effect has also been noted with finerenone, with higher doses causing a more significant reduction in urine albumin-creatinine ratios [70]. In the Finerenone in Reducing Cardiovascular Mortality and Morbidity in Diabetic Kidney Disease (FIGARO-DKD) trial, patients who were at high cardiovascular risk and had been excluded from the FIDELIO-DKD trial—those with stage 2 to 4 CKD and moderately elevated albuminuria; or state 1 or 2 CKD and severely increased albuminuria—were examined. Finerenone therapy improved cardiovascular outcomes as compared with placebo, primarily driven by a lower incidence of hospitalization for HF [71].

### 3.5. Inflammation

Patients with CKD have low-grade systemic inflammation, which has been associated with increased morbidity and mortality. The kidney microcirculation is tightly regulated, and inflammation alters the regulatory pathways and allows toxins like reactive oxygen species (ROS) to cause irreversible damage, nephron injury and failure, and lead to CKD [72]. The inflammatory molecules also cause a rise in fibroblast growth factor (FGF)-23 levels, an independent predictor of mortality in patients with CKD [73,74]. Similarly, inflammation has long been tied to cardiovascular disease by playing a central role in the pathogenesis of atherosclerotic plaque and is considered a significant cardiovascular risk factor [75,76]. In the Canakinumab Anti-Inflammatory Thrombosis Outcomes Study (CANTOS), IL-1β inhibition led to a significantly lower rate of recurrent cardiovascular events compared to the placebo group [77]. The RESCUE trial assessed the safety and efficacy of ziltivekimab, a human monoclonal antibody targeted against the Il-6 ligand, in patients with high cardiovascular risk, including those with chronic kidney disease. The data from the randomized, double-blinded trial demonstrated markedly reduced biomarkers (CRP, fibrinogen, serum amyloid A, haptoglobin, secretory phospholipase A2, and lipoprotein (a)), highlighting the need for further large-scale trials assessing the effects of anti-inflammatory medications in patients with CKD [78]. One such trial is underway: A Research Study to Look at How Ziltivekimab Works Compared to Placebo in People With Cardiovascular Disease, Chronic Kidney Disease, and Inflammation (ZEUS) (NCT05021835).

## Figures and Tables

**Table 1 jcm-12-06984-t001:** Epidemiology of CVD in DM with and without CKD and/or albuminuria.

Author	DM without CKD	DM with CKD	DM with Albuminuria
Rao and all (et al.) [6]	Death from vascular causes compared to non-DM, HR 2.32 (95% CI 2.11–2.56)		
Haffner et al. [8]	-Death from CVD in DM without MI compared to non-DM with prior MI, HR 1.2–1.4 (95% CI 0.7–2.6) -7-year incidence rates of MI in non-DM with and without prior MI: 18.8% and 3.5%, vs. 45.0% and 20.2%, (*p* < 0.001), in DM with and without prior MI respectively.		
Wannamethee et al. [9]	CVD and CV deaths: RR 2.82 (95% CI 1.85 to 4.28) in DM and 8.93 (95% CI 6.13 to 12.99) in patients with both DM and CHD.		
Zhao et al. [14]	CVD events in DM vs. non-DM vs. DM + CVD: HR 2.2 (95% CI 2.1–2.3), 2.9 (95% CI: 2.7–3.1) and 5.13 (95% CI: 4.7–5.5), respectively.		
CKD Prognosis consortium [17]		CV death: HR 1.52 for CKD3a, 2.4 for CKD3b and 13.5 for CKD4	CV death in CKD3a and 3b: HR 3·13 and 4.12 for ACR 30–299 mg/g, 4·97 and 6.10 for ACR > 300 mg/g
Branch et al. [19]	-ASCVD (non-fatal MI, non-fatal stroke, CVD death) in DM with CVD: HR 2.20 (1.92–2.53, *p* < 0.001) -All-cause mortality in DM with CVD: HR 1.29 (95% CI 1.51–2.12, *p* < 0.0001)	ASCVD in DM + CKD without CVD, HR 1.41 (95% CI 1.06–1.89, *p* = 0.02) ASCVD in DM + CKD + CVD: HR 2.35 (1.81–3.04), *p* < 0.001) -All-cause mortality in DM + CKD without CVD: HR 1.39 (1.01–1.90, *p* = 0.04) -All-cause mortality in DM + CKD + CVD: 2.36 (95% CI 1.75–3.13, *p* < 0.0001)	
Papademetriou et al. [20]		-ASCVD in CKD vs. non-CKD: HR 1.86 (95% CI 1.6–2.1), *p* < 0.001 -All-cause mortality in CKD vs. non-CKD: HR 1.97 (95% CI 1.70–2.28), *p* < 0.0001 -CV mortality in CKD vs. non-CKD: HR 2.18 (95% CI 1.75–2.72), *p* < 0.0001	
So WY et al. [22]		-CV end points across CKD stage 1–4: HR 1.00, 1.04, 1.05, and 3.23 respectively (*p* < 0.001) -All cause mortality across CKD stage 1–4: HR 1.00, 1.27, 2.34, and 9.82 respectively (*p* < 0.001)	
Drury et al. [23]		Total CVD events -eGFR 60–89 mL/min/1.73 m^2^: HR 1.14 (95% CI 1.01–1.29) -eGFR 30–59 mL/min/1.73 m^2^: HR 1.59 (95% CI 1.28–1.98) *p* < 0.001	CVD Risk in Type 2 DM with eGFR ≥ 90 mL/min/1.73 m^2^ -Microalbuminuria: HR 1.25 (95% CI 1.01–1.54) -Macroalbuminuria increased: HR 1.19 (95% CI 0.76–1.85),
Bruon et al. [24]		CV mortality compared to CKD1 across CKD stage 2–4: HR 0.65, 0.79, 0.67 and 2.03 (*p* = 0.27)	CV mortality in patient with AER 20–200 and >200ug/min: HR 1.06 and 2.0 respectively (*p* < 0.0001)
Targher at al [26]		All-cause and CV mortality per 1-SD decrease in eGFR: HR 1.53 (95% CI 1.2–2.0; *p* < 0.0001) and 1.51 (95% CI 1.05–2.2; *p* = 0.023), respectively.	All-cause and CV mortality per 1-SD increase in albuminuria: HR 1.14 (95% CI 1.01–1.3, *p* = 0.028) and 1.19 (95% CI 1.01–1.4, *p* = 0.043) respectively.
Ninomiya et al. [27]		CV events for every halving of baseline eGFR: HR 2.20 (95% CI 1.09 to 4.43)	CV events for every 10-fold increase in baseline UACR, HR 2.48 (95% CI 1.74–3.52)

Table: Chronic kidney disease (CKD), Diabetes Mellitus (DM), Hazard Ratio (HR), Cardiovascular (CV) Disease (CVD), Myocardial Infarction (MI), Confidence Interval (CI), Relative Risk (RR), Coronary Heart Disease (CHD), Confidence Interval (CI), Albumin-to-Creatinine Ration (ACR), Atherosclerotic Cardiovascular Disease (ASCVD), Estimated Glomerular Filtration Rate (eGFR), Albumin Excretion Rate (AER), Standard Deviation (SD), Urine albumin-creatinine ratio (UACR).

## Data Availability

Not available.

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
