# Peer review of "Cardiovascular Disease in Diabetes and Chronic Kidney Disease"

_jcm, 2023, doi:10.3390/jcm12226984_

Round 1

Reviewer 1 Report

Comments and Suggestions for Authors

Line 108 – This sentence is confusing and in disagreement with the cited reference. Please check.

Line 171 – Please rephrase for clarity.

Line 209 – The RENAAL study (N Engl J Med 2001; 345:861-869), published concomitantly with the Irbesartan Diabetic Nephropathy Trial, should also be cited.

There should be some reference to the fact that ACEI/ARB associations are not recommended.

Line 223 - Reference 38 focuses on inflammation, rather than on dyslipidemia or statin therapy.

Line 233 – Please rephrase

Line 285 – Reference 64 is in disagreement with the text in which it is quoted. Please check.

Comments on the Quality of English Language

 Extensive editing of English language required

Author Response

Reviewer 1:

Comments and Suggestions for Authors

Line 108 – This sentence is confusing and in disagreement with the cited reference. Please check.

Thank you for this comment. We have checked the original manuscript and have corrected to be more in line with the original publication:

“In the Action to Control Cardiovascular Risk in Diabetes (ACCORD) trial, the presence of CKD + DM was associated with a 41% increased risk (P = 0.02) for CVD composite primary outcome and 39% increased risk (P = 0.04) for all-cause mortality as compared to DM alone”

Line 171 – Please rephrase for clarity.

We have clarified this sentence further:

“Among patients with no CKD at baseline (eGFR >= 90 ml/min/1.73m2 and albumin-to-creatinine [ACR] ratio < 3.5 mg/mmol [<30 mg/gm]), worsening of ACR to >= 3.5 mg/mmol was associated with an increased risk for CVD; whereas a decrease in eGFR to 90 ml/min/1.73m² was not associated with an increased CVD risk[27] This suggests that at higher eGFR, albuminuria continues to carry important CVD prognostic information”

Line 229 – The RENAAL study (N Engl J Med 2001; 345:861-869), published concomitantly with the Irbesartan Diabetic Nephropathy Trial, should also be cited.

Thank you for pointing this out – we have corrected this sentence:

“Angiotensin receptor blockers (ARB) similarly showed effects in the Irbesartan Diabetic Nephropathy Trial (IDNT)  and in the Reduction of Endpoints in NIDDM with the Angiotensin II Antagonist Losartan Study (RENAAL), both of which demonstrated lower risk for progression to end stage renal disease (ESRD) and serum creatinine doubling time with the use of irbesartan or losartan, respectively”

There should be some reference to the fact that ACEI/ARB associations are not recommended.

This is an important point and thank you for the mention. We have added a sentence on line 238

“. On the other hand, clinical trial evidence has demonstrated the potential harm with combining ACEI and ARB for albuminuria reduction due to the risk of acute kidney injury and hyperkalemia .”

Line 223 - Reference 38 focuses on inflammation, rather than on dyslipidemia or statin therapy.

Thank you for this finding – we have updated the citation to reflect the importance of lipid lowering medication on patients with CKD and dyslipidemia.

Niki Katsiki, Dimitri P Mikhailidis & Maciej Banach (2019) Lipid-lowering agents for concurrent cardiovascular and chronic kidney disease, Expert Opinion on Pharmacotherapy, 20:16, 2007-2017, DOI: 10.1080/14656566.2019.1649394

Line 233 – Please rephrase

Thank you – we have improved the flow of the paragraph by amending the title of the section to include hypertension and albuminuria, and also rephrasing the sentence itself.

In addition, albuminuria has been identified as a risk factor for cardiac and kidney outcomes in patients with type 2 diabetes.

Line 285 – Reference 64 is in disagreement with the text in which it is quoted. Please check.

Thank you for pointing out this discrepancy. We have updated this section with a new reference, and new description of the inflammatory process as well.

Reviewer 2 Report

Comments and Suggestions for Authors

The article "Cardiovascular disease in diabetes and chronic kidney disease" provides a nice and concise review of the intersection between cardio-kidney-metabolic (CKM) conditions. I believe this article will serve as an excellent component of the special issue on CKD in diabetes.

I would like to offer the following comments for consideration by the authors:

General Comments

1. Consider use of person-first language throughout (e.g., substitute "person with diabetes" or "patient with diabetes" in place of "diabetic").

2. There are multiple acronyms thoughout the manuscript that are not defined and there are others that are defined multiple times. Recommend the authors review the paper and ensure all acronyms are defined upon first use and/or only defined once.

3. Recommend substituting the preferred term "kidney" for "renal" throughout where appropriate. This includes substituting ESKD for ESRD.

4. Use consistent terminology for "type 1 diabetes" and "type 2 diabetes" throughout the paper. A number of variations are used throughout (e.g., type 2 diabetes, DM type 2, type II DM).

Specific Comments

5. Abstract: "Kidney disease (KD)" is used in the abstract while "CKD" is used in the body of the manuscript. Recommend editing the abstract to be consistent with the body of the paper. 

6. Introduction: Recommend adding references to the introduction for data/statements made. 

7. Table: Recommend adding a table footnote to define acronyms within the table. 

8. Hypertension Section: Recommend substituting RAS for RAAS in this section since aldosterone antagonism is discussed independently later when discussing finerenone.

9. Glycemia and Glomerular Hyperfiltration Section: In my opinion the section discussing GLP-1RAs is a bit underdeveloped. While an exhaustive review of current data is not needed here, mention and reference of landmark CVOTs and key completed and ongoing trials in the setting of T2D and CKD (AWARD-7; FLOW) should be briefly mentioned.

10. Glycemia and Glomerular Hyperfiltration Section: Recommend updating reference #56 to the 2022 KDIGO recommendations and updating the stated recommendations for GLP-1 RA use here to align with the 2022 recs.

Comments on the Quality of English Language

Well written article. Per previous recommendation, the authors may consider editing to use person-first language throughout (e.g., avoid use of "diabetic").

Author Response

Reviewer 2:

Comments and Suggestions for Authors

The article "Cardiovascular disease in diabetes and chronic kidney disease" provides a nice and concise review of the intersection between cardio-kidney-metabolic (CKM) conditions. I believe this article will serve as an excellent component of the special issue on CKD in diabetes.

I would like to offer the following comments for consideration by the authors:

General Comments

  1. Consider use of person-first language throughout (e.g., substitute "person with diabetes" or "patient with diabetes" in place of "diabetic").

Thank you for this important reminder to utilize person-first language. We have made every attempt to capture and modify these in our paper.

  1. There are multiple acronyms throughout the manuscript that are not defined and there are others that are defined multiple times. Recommend the authors review the paper and ensure all acronyms are defined upon first use and/or only defined once.

Thank you for this comment – we have gone through and ensured that acronyms are defined once and then used consistently throughout. 

  1. Recommend substituting the preferred term "kidney" for "renal" throughout where appropriate. This includes substituting ESKD for ESRD.

Thank you for this important point. We have used the term kidney instead of renal, unless there was as specific reference to a study that used the term renal (for instance as an outcome). 

  1. Use consistent terminology for "type 1 diabetes" and "type 2 diabetes" throughout the paper. A number of variations are used throughout (e.g., type 2 diabetes, DM type 2, type II DM).

We have adjusted the terminology to ensure that it is consistent throughout the paper (type 1 and type 2 DM). 

Specific Comments

  1. Abstract: "Kidney disease (KD)" is used in the abstract while "CKD" is used in the body of the manuscript. Recommend editing the abstract to be consistent with the body of the paper. 

Thank you – we have used the term CKD throughout.

  1. Introduction: Recommend adding references to the introduction for data/statements made. 

We have added references to the statements made. 

  1. Table: Recommend adding a table footnote to define acronyms within the table. 

Thank you, updated as recommended

  1. Hypertension Section: Recommend substituting RAS for RAAS in this section since aldosterone antagonism is discussed independently later when discussing finerenone.

Thank you we have substituted RAS in the hypertension section

  1. Glycemia and Glomerular Hyperfiltration Section: In my opinion the section discussing GLP-1RAs is a bit underdeveloped. While an exhaustive review of current data is not needed here, mention and reference of landmark CVOTs and key completed and ongoing trials in the setting of T2D and CKD (AWARD-7; FLOW) should be briefly mentioned.

Thank you for this important suggestion. We have added sentences discussing these important CVOTs of GLP-1Ras.   (lines 254 – 273 in the markup).

  1. Glycemia and Glomerular Hyperfiltration Section: Recommend updating reference #56 to the 2022 KDIGO recommendations and updating the stated recommendations for GLP-1 RA use here to align with the 2022 recs.

We have updated the reference as suggested. 
